# Oral Immunization with Attenuated *Salmonella* Choleraesuis Expressing the FedF Antigens Protects Mice against the Shiga-Toxin-Producing *Escherichia coli* Challenge

**DOI:** 10.3390/biom13121726

**Published:** 2023-11-30

**Authors:** Guihua Zhang, Yang Fu, Yu’an Li, Quan Li, Shifeng Wang, Huoying Shi

**Affiliations:** 1College of Veterinary Medicine, Yangzhou University, Yangzhou 225009, China; dz120180002@stu.yzu.edu.cn (G.Z.); mz120191098@yzu.edu.cn (Y.F.); liquan2018@yzu.edu.cn (Q.L.); 2Jiangsu Co-Innovation Center for the Prevention and Control of Important Animal Infectious Diseases and Zoonoses, Yangzhou 225009, China; 3Department of Infectious Diseases and Immunology, College of Veterinary Medicine, University of Florida, Gainesville, FL 32611-0880, USA; shifengwang@ufl.edu; 4Joint International Research Laboratory of Agriculture and Agri-Product Safety, Yangzhou University (JIRLAAPS), Yangzhou 225009, China

**Keywords:** edema disease, recombinant attenuated *Salmonella* vaccine, Shiga toxin producing *Escherichia coli*, FedF protein, rStx2eA protein, vaccine

## Abstract

Edema disease (ED) is a severe and lethal infectious ailment in swine, stemming from Shiga-toxin-producing *Escherichia coli* (STEC). An efficient, user-friendly, and safe vaccine against ED is urgently required to improve animal welfare and decrease antibiotic consumption. Recombinant attenuated *Salmonella* vaccines (RASV) administered orally induce both humoral and mucosal immune responses to the immunizing antigen. Their potential for inducing protective immunity against ED is significant through the delivery of STEC antigens. rSC0016 represents an enhanced recombinant attenuated vaccine vector designed for *Salmonella* enterica serotype Choleraesuis. It combines sopB mutations with a regulated delay system to strike a well-balanced equilibrium between host safety and immunogenicity. We generated recombinant vaccine strains, namely rSC0016 (pS-FedF) and rSC0016 (pS-rStx2eA), and assessed their safety and immunogenicity in vivo. The findings demonstrated that the mouse models immunized with rSC0016 (pS-FedF) and rSC0016 (pS-rStx2eA) generated substantial IgG antibody responses to FedF and rStx2eA, while also provoking robust mucosal and cellular immune responses against both FedF and rStx2eA. The protective impact of rSC0016 (pS-FedF) against Shiga-toxin-producing *Escherichia coli* surpassed that of rSC0016 (pS-rStx2eA), with percentages of 83.3%. These findings underscore that FedF has greater suitability for vaccine delivery via recombinant attenuated *Salmonella* vaccines (RASVs). Overall, this study provides a promising candidate vaccine for infection with STEC.

## 1. Introduction

ED is a condition of intestinal toxemia triggered by STEC, often seen in piglets aged 4–12 weeks [1]. Clinical presentations include eyelid swelling, paralysis, abnormal vocalizations, neurological signs, and a significantly increased mortality rate, leading to substantial economic losses in the swine farming industry [2]. In the realm of clinical application, the pathogenic strain of *Escherichia coli* has been undergoing evolutionary changes [3]. Moreover, the emergence of novel resistance genes, fueled by antibiotic misuse, has given rise to widespread multidrug resistance. This complex scenario engenders significant impediments to the efficacious management and containment of Escherichia-coli-associated maladies [4]. Consequently, there persists an exigent demand for the expeditious development of a novel vaccine technology platform [5,6]. Meanwhile, *Salmonella* Choleraesuis is a significant pathogen responsible for paratyphoid fever in piglets aged 2 to 4 months. This bacterium can induce widespread illness in recently weaned piglets, resulting in various clinical symptoms such as sepsis and localized inflammation in other tissues. Its impact on the breeding industry is pivotal [7,8]. STEC and *Salmonella*, both Gram-negative facultative anaerobic bacilli, are prevalent commensal and pathogenic bacteria within the gastrointestinal tracts of warm-blooded animals [9].

The Recombinant Attenuated *Salmonella* Vaccines (RASVs) approach stands out as a compelling platform for delivering antigens [10]. It offers an economical and needle-free approach to transporting foreign antigens, leading to a substantial enhancement in vaccine immunogenicity and cost-effectiveness [11]. To date, RASVs have effectively delivered antigens from various sources, including bacteria, viruses, and parasites, eliciting immune responses [12,13,14]. Many research efforts have utilized attenuated *Salmonella* as a vehicle for delivering *E. coli* antigens, with the goal of preventing and managing pathogenic Escherichia coli. Up to now, the researchers have used attenuated *Salmonella* as a vector to express the pathogenic *Escherichia coli* antigens K88, K99, FedA, FedF, FasA, and F41 [15,16,17,18,19,20,21,22]. *Salmonella* possesses inherent attributes as a carrier. These encompass its notable adjuvant properties and the capability to produce various Toll-like receptor agonists like flagellin, lipopolysaccharides, and lipoproteins. These components serve as potent adjuvants, enhancing the generated immune response. This significantly boosts both the Th1-dominant and mucosal immune response to exogenous antigens [23,24,25]. RASVs could use a type III secretion system (T3SS) for injection of effector proteins into the host cell cytosol which presented by MHC-I molecules generating efficient CD8^+^ T-cell responses [12,26,27,28]. RASVs are recognized for their capacity to vigorously stimulate both the humoral and cellular components of the immune response in vaccinated individuals. They can grow within the host’s body and have been extensively employed in the management of *Salmonellosis*. Their capacity to access the host effectively through mass oral administration along the mucosal route results in comprehensive protection against *Salmonella* [10]. Research has demonstrated that the utilization of RASVs carrying exogenous antigens can confer dual protection at the same time [29,30,31].

The selection of antigens has always been a research focus in the development of edema disease vaccines. Among them, F18 fimbriae and Stx2e are highly concerned [32,33]. The clinical evidence has substantiated a frequent correlation between F18 fimbriae and piglet diarrhea. The F18-fimbriae-fed gene cluster encompasses essential genes including *fedA* (coding for the primary subunit protein), *fedB* (coding for the molecular chaperone), *fedC* (coding for the introducer protein), *fedE* (coding for the secondary subunit), and *fedF* (coding for the adhesin) [34]. It has been observed that the gene *fedF*, responsible for encoding the adhesion subunit, demonstrates noteworthy conservation [35]. Moreover, in vitro experiments have confirmed that mutant strains lacking the FedF gene experience a decline in adhesion capability [36]. This has led researchers to predominantly target FedF for vaccine development purposes. As for the edema-disease-associated Stx2e whole toxin, it consists of a toxic A subunit housing N-terminal glycosidase activity and five nontoxic B subunits responsible for cell receptor binding [37,38,39]. Notably, the A subunit assumes the role of toxicity induction. In consideration of this, during the formulation of vaccines directed at A-subunit proteins, the essential step of codon optimization becomes imperative to effectively mitigate and eliminate toxicity [40].

In order to create a potent edema disease vaccine, we utilized a recombinant strain known as rSC0016 [41]. Vaccine candidate strains, namely rSC0016(pS-FedF) and rSC0016(pS-rStx2eA), were engineered to express the FedF and rStx2eA antigens. We assessed the immune responses elicited by rSC0016(pS-FedF) and rSC0016(pS-rStx2eA), along with their protective efficacy against STEC, using a mouse model. Our findings demonstrated that these constructs might present a novel avenue in the pursuit of preventing and controlling edema disease.

## 2. Materials and Methods

### 2.1. Animals and Ethics Statement

Female BALB/c mice were procured from the Comparative Medicine Center at Yangzhou University in Jiangsu, China. All animal experiments adhered rigorously to the animal welfare regulations outlined in the Animal Research Committee Guidelines of Jiangsu Province (License Number: SYXK(SU) 2017-0044) and received approval from the Ethics Committee for Animal Experimentation at Yangzhou University. In the course of the animal experiments, every endeavor was made to reduce suffering and optimize animal welfare.

### 2.2. Plasmids and Bacterial Strains

The strains and plasmids utilized in this study are presented in Table 1. The STEC strain STEC20, preserved in our laboratory, was used to amplify the gene fragments *fedf* and *rstx2eA*. Plasmid pYA3493 functions as an Asd+ vector, and plasmids pS-FedF and pS-rStx2eA, derived from pYA3493, carry the *fedF* or *rstx2eA* gene from STEC20, respectively. The strain rSC0016 was prepared through prior laboratory research [41].

### 2.3. Protein Expression, Protein Purification, and Antibody Preparation

The sequences of *fedF* or *rstx2eA* genes were amplified via PCR and then inserted into the expression vector pET28a. For rStx2eA amplification, overlap PCR was employed to substitute the codons at the 167th and 170th amino acid positions with codons encoding Gln and Lys. This modification aimed to reduce its toxicity and enhance its immunogenicity [40]. The primers utilized in this study are detailed in Table 2. The vectors pET28a-FedF and pET28a-rStx2eA transformed *E. coli* BL21 (DE3) competent cells to generate purified proteins. BL21 cells harboring pET28a-FedF and pET28a-rStx2eA were grown in LB medium. The medium was supplemented with kanamycin, and the cells were incubated at 37 °C. The incubation continued until they reached an OD_600_ of 0.6, which marked the logarithmic growth phase. Following this, the bacterium was subjected to induction for a duration of 4 h using IPTG. The proteins were then purified utilizing Ni-NTA. Purified recombinant proteins were measured for their protein concentrations using the BCA method, identifying proteins through Western blot analysis with anti-His-tag monoclonal primary antibodies (Boster Biological Technology Co., Ltd., Wuhan, China).

The proteins were diluted as necessary and mixed with an equal volume of QuickAntibody-Mouse3W adjuvant (Biodragon, Suzhou, China). Six-week-old female BALB/c mice received two intramuscular immunizations in the leg, spaced two weeks apart, with each mouse receiving 20 μg of the immunogen ever time. One week after the final immunization, blood samples were collected from both immunized and nonimmunized mice. Following centrifugation at 3000× *g* for 15 min, the sera were separated, and their antibody titers were evaluated using ELISA assay.

### 2.4. Indirect ELISA

An ELISA analysis was conducted to determine antibody titers targeting FedF and rStx2eA as previously described [41]. Recombinant FedF and rStx2eA proteins or *S.* Choleraesuis OMPs (0.5 μg/mL) were immobilized onto microtiter plates using 0.1 M sodium carbonate buffer (pH 9.6), subsequent to an overnight incubation at 4 °C. Following the incubation with a blocking buffer, the wells were subjected to a 2 h incubation at 37 °C with polyclonal antibody serum that had been appropriately diluted in PBST (varying from 1:1000 to 1:128,000) or serum and vaginal mucosal flushing solution from *Salmonella*-vaccine-immunized subjects (ranging from 1:100 to 1:12,800). Following that, 100 μL of goat anti-mouse IgG or goat anti-mouse IgA antibody (1:5000) was allowed to incubate at 37 °C for 90 min. The color reaction was initiated by the addition of 100 μL of TMB (Solarbio, Beijing, China) and allowed to progress for 15 min. The reaction was subsequently terminated with the addition of 50 μL of 2 M H_2_SO_4_. Lastly, the optical density (OD) was assessed at 450 nm using an automated microplate reader. The outer membrane proteins (OMPs) from the wild-type *S.* Choleraesuis strain C78-3 were obtained using the established procedure as previously described [41]. In brief, bacterial pellets were collected by centrifugation and suspended in a 4 mL buffer comprising 1% Sarkosyl and 20 mM of Tris-HCl (pH 8.6). The suspension was then incubated on ice for 30 min. Subsequently, OMPs were isolated by centrifugation at 4 °C for 1 h at 132,000× *g*. The separated OMPs were resuspended in a 4 mL buffer containing 20 mM of Tris-HCl (pH 8.6).

### 2.5. Construction of Vaccine Strains and Detection of Proteins Expression

The genes *fedF* and *rstx2eA* were gain from pET28a-FedF and pET28a-rStx2eA and integrated into the *EcoR* Ι and *Hind* ΙΙΙ restriction enzyme sites of the plasmid pYA3493 backbone, resulting in plasmids named pS-FedF and pS-rStx2eA, respectively. The primer sequences used in this study can be found in Table 2. The vector control plasmid pYA3493, along with the pS-FedF and pS-rStx2eA plasmids, were introduced into the *asd*-deficient *S.* Choleraesuis vector rSC0016, resulting in strains designated as rSC0016(pYA3493), rSC0016(pS-FedF), and rSC0016(pS-rStx2eA). To confirm the successful expression of FedF and rStx2eA proteins in these vaccine candidate strains, Western blot was conducted using the anti-FedF and anti-rStx2eA serum prepared earlier.

### 2.6. Bacterial Growth Curves

Cultures of rSC0016(pS-FedF), rSC0016(pS-rStx2eA), and rSC0016(pYA3493) in the mid-exponential growth phase were adjusted to an OD_600_ of 0.5. They were then diluted 1:100 in fresh LB medium and subsequently incubated at 37 °C. Bacterial growth curves were derived from OD_600_ measurements taken every two hours over an 8 h period.

### 2.7. Immunization in Mice

The bacterial solution preserved at −80 °C was revived on an LB plate enriched with 0.2% arabinose and mannose. Subsequently, individual colonies were transferred into LB liquid medium supplemented with 0.2% arabinose and mannose, and they were incubated at 37 °C for 16–18 h. For the inoculation, a 1:100 dilution was made in LB liquid medium enriched with 0.2% arabinose and mannose. The mixture was shaken and cultured on a constant temperature shaker at 37 °C until the OD_600_ value of the bacterial solution reached approximately 0.9. The bacterial solution was subsequently centrifuged, washed with sterile PBS, and the bacterial pellet was collected. Afterwards, PBS was added to resuspend the bacterial pellet, resulting in a thoroughly mixed immune bacterial solution [43].

Immunoprotective experiments were conducted with 6-week-old female BALB/c mice (*n* = 9). Mice were kept 1 week after arrival to acclimate them to our animal facility before immunization and were deprived of food and water for 6 h before oral immunization. Two groups received PBS through oral administration, serving as healthy control and subsequent challenge control groups. Furthermore, two groups were subjected to oral pipette feeding with a 20 μL (1 ± 0.3 × 10^9^ CFU) bacterial solution of rSC0016 (pS-FedF) and rSC0016 (pS-rStx2eA), while a separate group received the bacterial solution of rSC0016 (pYA3493) as an empty control. Food and water were returned to the mice 30 min after immunization. After 21 days from the initial immunization, each group received an additional immunization. On the 21st and 35th days after the initial immunization, serum samples were collected to assess IgG levels. In addition, vaginal mucosal flushing solutions were obtained by rinsing the vagina with sterile PBS to measure secretory IgA levels. The collected blood was placed in a refrigerator set at 4 °C overnight, followed by centrifugation to extract serum. All collected serum and vaginal rinse samples from each group were stored at −80 °C for preservation. Specific antibody titers in both vaginal mucosal flushing solution and serum were detected using an indirect ELISA. IFN-γ and IL-4 levels were detected using the Mouse IFN-γ and IL-4 ELISA KIT (Beijing Solarbio Science & Technology Co., Ltd., Beijing, China), following the provided instructions.

### 2.8. Challenge in Mice

Select a single colony of STEC20 and allow it to incubate overnight in 5 mL of LB liquid culture medium. On the following day, inoculate the colony at a 1:100 ratio into 50 mL of LB liquid medium and shake it at 37 °C for cultivation. When the OD_600_ value of the bacterial solution reaches 0.8, harvest the bacterial cells. Resuspend the bacterial cells in 300 μL of PBS. Perform successive 10-fold dilutions and select the appropriate dilution for the challenge. Randomly divide 25 female BALB/c mice into 5 groups, each containing 5 mice. Inject the leg muscles of each mouse in the groups with 4 different target dilutions of the bacterial solution. An additional 5 mice comprise the blank control group, receiving individual injections of the same volume of PBS. Utilize the Reed–Muench method to calculate the LD_50_ based on these injections shown in Appendix A. Three weeks after the second immunization, inject STEC20 into their leg muscles for challenge, using a challenge dose equivalent to 3.5 times the LD_50_. Following the challenge, continuously observe the mice to calculate the survival rate based on their mortality status. The experimental design is shown in Appendix A.

### 2.9. Statistical Analysis

Statistical analyses were conducted using GraphPad Prism 8. Data were presented as the mean ± SEM for all assays. Group comparisons were performed using the Mann–Whitney U Test. A *p*-value of less than 0.05 was considered statistically significant for all tests.

## 3. Results

### 3.1. Expression Recombinant FedF and rStx2eA Proteins and Production Polyclonal Antibody Sera

Using the STEC20 strain as a template, we amplified an 840 bp *fedF* gene fragment and an 891 bp *rstx2eA* gene fragment through PCR (Figure 1A,B). The *fedF* and *rstx2eA* fragments were then inserted into the pET28a vector. Positive plasmids were identified using double-restriction endonuclease digestion (Figure 1C). Afterward, they were introduced into the expression strain *E. coli* BL21(DE3), resulting in the creation of BL21(pET28a-rStx2eA) and BL21(pET28a-FedF) strains. Western blot showed that both BL21(pET28a-rStx2eA) and BL21 (pET28a-FedF) lanes exhibited specific bands of the expected size, while the empty control strain did not show any bands (Figure 1D). The results demonstrated the successful expression of FedF and rStx2eA proteins by BL21(pET28a-FedF) and BL21(pET28a-rStx2eA), respectively. Subsequently, the purified FedF protein and rStx2eA protein were used to generate polyclonal antibody sera in mice. Antibody titers were determined via indirect ELISA, revealing serum titers of 1:51,200 for the FedF antigen and 1:25,600 for the rStx2eA antigen.

### 3.2. Construction and Characterization of rSC0016(pS-FedF) and rSC0016(pS-rStx2eA)

The *fedF* and *rstx2eA* genes from STEC20 were inserted into the pYA3493, resulting in the creation of pS-FedF and pS-rStx2eA (Figure 2A). The pS-FedF and pS-rStx2eA plasmids were verified using double-restriction enzyme digestion. The sizes of the fragments were as follows: 3113 bp for pYA3493, 840 bp for *fedF*, and 891 bp for *rstx2eA* (Figure 2B). The pS-FedF and pS-rStx2eA plasmids were introduced into the competent rSC0016 strain to create vaccines rSC0016(pS-rStx2eA) and rSC0016(pS-FedF). Equal volumes of bacterial solutions of three vaccine strains were subjected to Western blot analysis. The results revealed an approximately 36 kDa protein expression in rSC0016(pS-FedF) and an approximately 38 kDa protein expression in rSC0016(pS-rStx2eA). No distinct bands were observed in the lane corresponding to rSC0016(pYA3493) (Figure 2C). These observed protein band sizes aligned with the anticipated sizes of FedF and rStx2eA proteins, indicating the accurate synthesis of the target antigens by both vaccine candidates. The growth curve outcomes indicated that despite carrying heterologous antigens, there was no notable difference in the growth rate among rSC0016(pS-FedF), rSC0016(pS-rStx2eA), and rSC0016(pYA3493) (Figure 2D).

### 3.3. S. Choleraesuis Vaccine Vector Strains rSC0016(pS-FedF) and rSC0016(pS-rStx2eA) Elicited Elevated Serum IgG and Mucosal IgA Responses to FedF and rStx2eA

Indirect ELISA measured IgG levels for FedF and rStx2eA proteins in serum after 3 and 5 weeks of initial immunization, along with IgA levels in vaginal rinses. Also, IgG levels for induced C78-3 outer membrane proteins (OMPs) were assessed using indirect ELISA. In comparison to the rSC0016(pYA3493) and the blank control group, the results revealed that following the first and second immunizations, those immunized with rSC0016(pS-FedF) and rSC0016(pS-rStx2eA) demonstrated markedly elevated concentrations of serum IgG and mucosal IgA against FedF and rStx2eA. Moreover, all immunized groups exhibited higher antibody levels at 5 weeks post-initial immunization compared to 3 weeks (Figure 3A,B). There was no significant difference in antibody levels between the rSC0016(pS-FedF) immune group and the rSC0016(pS-rStx2eA) immune group. The rSC0016(pYA3493) empty vector group did not express heterologous antigens; consequently, no corresponding antibodies were produced. Notably, the antibody levels against the OMPs induced by rSC0016(pS-FedF) and rSC0016(pS-rStx2eA) were comparable to those of the empty vector immunized group after both immunizations (Figure 3C). This indicates that the two vaccine candidate strains not only induced immune responses to heterologous antigens but also triggered immune responses against Salmonella. In contrast, the blank control group did not generate any antibodies.

### 3.4. S. Choleraesuis Vaccine Vector Strains rSC0016(pS-FedF) and rSC0016(pS-rStx2eA) Induced Higher Levels of IFN-γ and IL-4 in Mice

Seven days after the second immunization, three mice were randomly selected from each group, and their spleens were obtained and homogenized. Following multiple freeze–thaw cycles, the samples were subjected to centrifugation at 12,000× *g* for 1 min to acquire the supernatant. This supernatant was then used for subsequent analysis. Using the obtained supernatant, cytokine levels were measured using IL-4 and IFN-γ ELISA kits. In comparison to the rSC0016(pYA3493) group, both vaccine formulations induced significantly higher levels of IL-4 and IFN-γ in immunized groups. The spleen samples from rSC0016(pS-FedF) mice exhibited an average IFN level of 600 pg/mL, which is five times higher than that of the rSC0016(pYA3493) group. Likewise, the rSC0016(pS-FedF) group exhibited IL-4 levels approximately three times greater than the empty vector group. Furthermore, there was no significant disparity in IL-4 and IFN-γ cytokine levels between the spleens of mice immunized with the rSC0016(pS-rStx2eA) strain and those immunized with the rSC0016(pS-rStx2eA) strain. The blank control group did not show detectable serum factor levels (Figure 4).

### 3.5. S. rSC0016(pS-FedF) and rSC0016(pS-rStx2eA) Vaccine Strains Protects Mice against STEC Infection

Except for the mice in the PBS group, all other mice were challenged by injecting STEC20 into their leg muscles at a dose equivalent to 3.5 times the LD_50_. Within a span of 36 h following exposure, mice in both the rSC0016(pS-FedF) immunized group and the blank control group exhibited gradual fatalities. The survival rates for the rSC0016(pS-FedF) and rSC0016(pS-rStx2eA) immunization groups were 83.3% and 33.3%, respectively (Figure 5). The surviving mice displayed mild clinical symptoms including mental fatigue, disordered hair, and eyelid congestion. They subsequently resumed normal activities and eating patterns. The results underscore that both the rSC0016(pS-FedF) and rSC0016(pS-rStx2eA) immunization groups conferred a certain level of protection against Escherichia coli infection in mice. Notably, the protective efficacy of rSC0016(pS-FedF) was significantly superior in comparison to that of rSC0016(pS-rStx2eA).

## 4. Discussion

STEC is a pathogen of edema disease (ED). Being a highly deadly infectious disease among piglets, it has resulted in substantial economic losses to the global breeding industry. Vaccine immunization remains a powerful measure for preventing and controlling edema disease [5]. Given the rise of multidrug-resistant strains of STEC in afflicted pigs, employing vaccines remains a potent strategy for preventing and managing edema disease [5,44]. Inactivated vaccines are currently widely used in the market, but they require multiple immunizations and large doses, thus increasing the cost of use. Consequently, there persists a necessity for the development of vaccines that are both more efficient and user-friendly, while ensuring safety [45].

Our recent research has devised mechanisms for controlled delayed attenuation and antigen synthesis [41,46,47]. During the initial phases of oral immunization, the meticulously controlled delayed attenuated *Salmonella* vaccine strain demonstrates efficient colonization of deep lymphoid tissues similar to virulent wild-type strains. Subsequently, due to the lack of mannose and arabinose in the host, the rSC0016 demonstrate detoxification characteristics and do not elicit disease symptoms. Research indicates that the controlled delayed antigen synthesis system can govern the production of foreign antigens. This empowers vaccines to induce robust levels of specific antigen antibodies upon colonization of lymphatic tissue [48]. FedF is regarded as a promising protective antigen for vaccine development. This is because it is relatively conserved and associated with bacterial adhesion [35,36]. Active immunity to Stx2e toxin induces strong immune responses in piglets and sows [49].

Vaccine candidate strains rSC0016(pS-FedF) and rSC0016(pS-rStx2eA) were prepared that can express two virulence-related factors FedF and rStx2eA (including dual mutations) of STEC. To assess the potential of rSC0016(pS-FedF) and rSC0016(pS-rStx2eA) as vaccine candidates against STEC, we analyzed the characteristics of these strains. The growth patterns of rSC0016(pS-FedF) and rSC0016(pS-rStx2eA) exhibited a similarity to that of the control strain containing an empty vector, which is rSC0016(pYA3496). Furthermore, the production of exogenous protective antigens FedF and rStx2eA was observed in rSC0016(pS-FedF) and rSC0016(pS-rStx2eA). Following the immunization of mice, the vaccine strains elicited a robust, targeted immune response, resulting in elevated titers of IgG and IgA. Th1 cells are pivotal in orchestrating cellular immune responses against intracellular parasites [50,51], mainly secreting IFN-γ. IL-4 is secreted by Th2-type cells, and its primary function is to stimulate B-cell proliferation. It plays a significant role in both humoral immune responses and mechanisms of inflammation [52]. In this research, the average cytokine levels of IL-4 and IFN-γ in the spleens of mice immunized with rSC0016(pS-FedF) and rSC0016(pS-rStx2eA) were greater than those in the control group. Furthermore, the immune groups involving rSC0016(pS-FedF) and rSC0016(pS-rStx2eA) also provoked comparable levels of IL-4 and IFN-γ, suggesting the establishment of a harmonious Th1/Th2 immune response.

As for the protective effect, our results show that the rSC0016(pS-FedF) group achieved an 83.3% post-challenge survival rate. This outcome aligns closely with the survival rate of approximately 80%, as reported by Ren W et al. in their study involving a vaccine targeting the pili adhesion factor FedF [53]. These results further underscore the potential of the pili subunit FedF as a prime target in the development of edema disease vaccines. In contrast to the impressive protective efficacy observed with the modified Stx2e whole-toxin subunit vaccine [49], the protective effectiveness achieved by utilizing the Salmonella vector to deliver rStx2eA post-immunization yielded suboptimal results in this investigation. This discrepancy could be attributed to the apparent lack of production of efficacious neutralizing antibodies following immunization with rSC0016(pS-rStx2eA) in mice [54]. Additionally, a limitation of this study is the absence of a comparative analysis between the immune efficacies of the vaccine strains and commercially available edema disease vaccines.

Overall, rSC0016(pS-FedF) and rSC0016(pS-rStx2eA) vaccines induce cellular, mucosal, and humoral immune responses in mice. The rSC0016(pS-FedF) amalgamates the benefits of both rSC0016 and FedF, striking a favorable equilibrium between host safety and immunogenicity, thereby providing protection against SETC in mice. The findings of this research strongly indicate that the rSC0016(pS-FedF) strain holds significant promise as a candidate for the development of vaccines targeting Shiga-toxin-producing *Escherichia coli*.

## Figures and Tables

**Figure 1 biomolecules-13-01726-f001:**
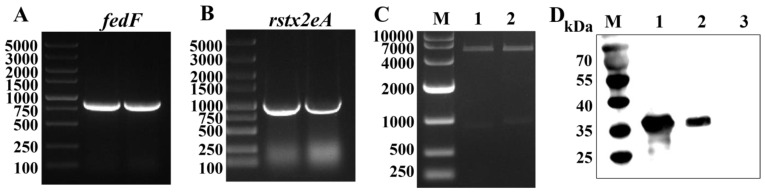
Amplification, clone, and expression of FedF and Stx2eA. (**A**) Amplification of *fedF* gene. M: DL 5000 DNA Marker. (**B**) Amplification of *rstx2eA* gene. M: DL 5000 DNA Marker. (**C**) Identification of recombinant plasmids by restriction endonuclease digestion. M: DL 10,000 DNA Marker; 1: pET28a-FedF; 2: pET28a-Stx2eA. (**D**) The results of Western blot for FedF and Stx2eA in BL21. M: Protein Marker. 1: BL21(pET28a-FedF); 2: BL21(pET28a-Stx2eA); 3: BL21(pET28a). Original images of (**A**–**D**) can be found in Appendix A.

**Figure 2 biomolecules-13-01726-f002:**
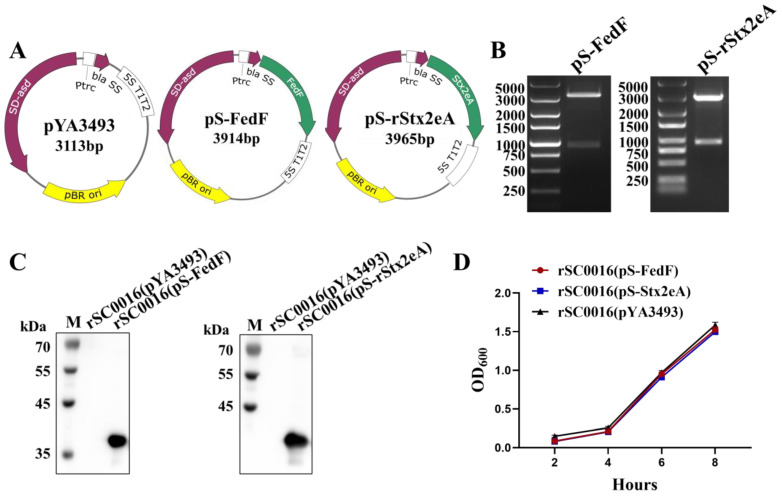
Plasmid maps and construction results, phenotypic characteristics of vaccine candidate strains. (**A**) Plasmid maps of pYA3493, pS-FedF, and pS-Stx2eA. (**B**) Identification of recombinant plasmid pS-FedF and pS-Stx2eA by restriction enzyme digestion. *EcoR*-I- and *Hind*-ΙΙΙ-digested pS-FedF and pS-Stx2eA. (**C**) The expression of FedF and Stx2eA in rSC0016 was analyzed by Western blot. (**D**) Growth curves of the rSC0016(pS-FedF), rSC0016(pS-Stx2eA), and rSC0016(pYA3493) strains in the LB medium. Original images of (**B**,**C**) can be found in Appendix A.

**Figure 3 biomolecules-13-01726-f003:**
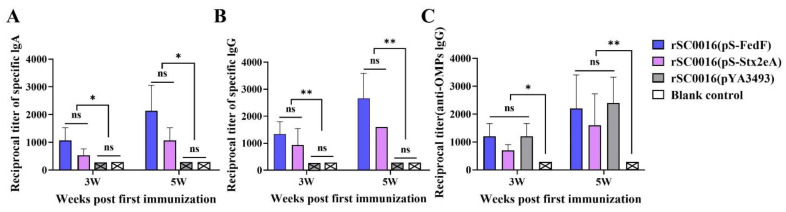
Detection of antibody titer in the immunized mice. (**A**) FedF-specific and Stx2eA-specific IgG antibody titer in serum determined by ELISA. (**B**) FedF-specific and Stx2eA-specific IgA antibody titer in vaginal rinses determined by ELISA. (**C**) OMPs-specific IgG antibody titer in serum determined by ELISA. The results are expressed as the mean ± SD. Degrees of significance are indicated as follows: * *p* <  0.05; ** *p*  <  0.01; ns *p* ≥ 0.05.

**Figure 4 biomolecules-13-01726-f004:**
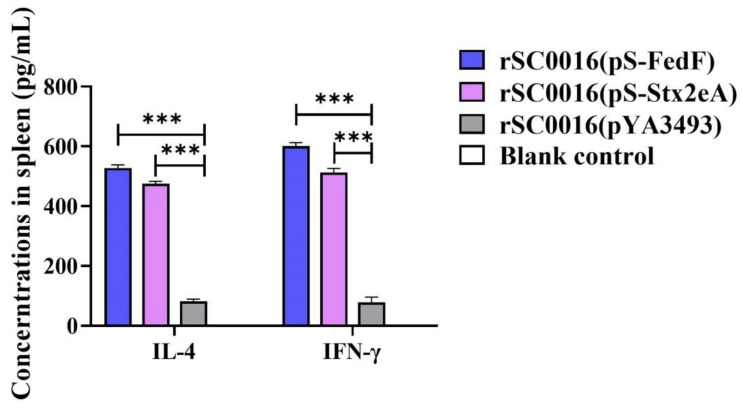
Levels of secreted IL-4 and IFN-γ were assayed by ELISA. Splenic lymphocytes were used to evaluate cytokine secretion in vitro following restimulation with purified FedF and Stx2eA protein, respectively. The results are expressed as the mean ± SD. Degrees of significance are indicated as follows: *** *p* < 0.01.

**Figure 5 biomolecules-13-01726-f005:**
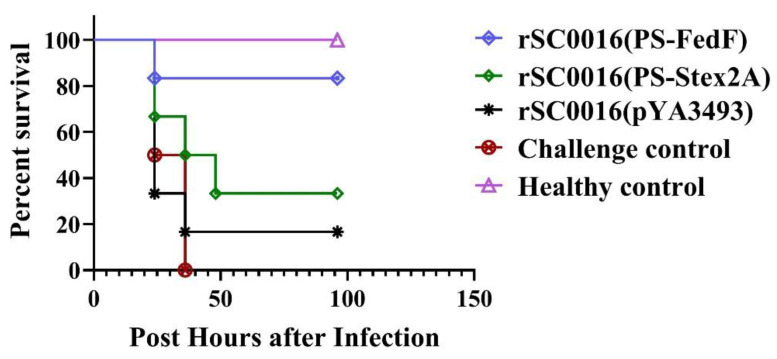
Protective efficacy of developed vaccine. Survival rates of mice after the Shiga-toxin-producing *Escherichia coli* challenge were determined.

**Table 1 biomolecules-13-01726-t001:** Bacterial strains and plasmids used in this study.

Strains and Plasmid	Characteristics	Sources, References, or Function
*E. coli* strains		
DH5α	For amplification of the recombinant plasmid	
BL21	For expression of the recombinant plasmids	Invitrogen
χ7213	thi-1, thr-1, leuB6, fhuA21, lacY1, glnV44, asdA4, recA1, RP4 2-Tc::Mu pir; Kmr	Provided by Dr. Roy Curtiss III
*S.* Choleraesuis		
C78-3	Wild-type, virulent, CVCC79103	China Institute of Veterinary Drugs Control
rSC0016	ΔPcrp527::TT araC PBADcrpΔpmi-2426ΔrelA199::araC PBADlacI TTΔsopB1686 ΔasdA33	[41]
Shiga-toxin-producing *Escherichia col*i STEC20	Wild-type, virulent	[42]
Plasmids		
pYA3493	Plasmid Asd+; pBR ori, β-lactamase signal sequence-based periplasmic secretion plasmid	Provided by Dr. Roy Curtiss III
pET28a	Expression vector, Kanr	Novagen
pMD19-T	Cloning vector; Ampr	TaKaRa
pET28a-FedF	A recombinant expression vector containing FedF; Kanr	This study
pET28a-rStx2eA	A recombinant expression vector containing rStx2eA; Kanr	This study
pS-FedF	pYA3493 with FedF	This study
pS-rStx2eA	pYA3493 with rStx2eA	This study

Kanr, Kanamycin resistance; Ampr, Ampicillin resistance.

**Table 2 biomolecules-13-01726-t002:** The primers information.

Primer Name	Sequences (5′-3′)	References
fedF-28a-F	CCGGAATTCACTCTACAAGTAGACAAGTCTGTT	This study
fedF-28a-R	CCCAAGCTTTTACTGTATCTCGAAAACAAT	
stx2eA-28a-1	CCGGAATTCCAGGAGTTTACGATAGACT	This study
stx2eA-28a-2	TATTTGCCTGAACTTTAAGGCTTGTGCTGTGACAGTGACAAAACG	
stx2eA-28a-3	CGTTTTGTCACTGTCACAGCACAAGCCTTAAAGTTCAGGCAAATA	
stx2eA-28a-4	CCCAAGCTTTTATTCACCAGTTGTATATAAAGG	
pYA3493-F	AACGCTGGTGAAAGTAAAAGATG	This study
pYA3493-R	CAGACCGCTTCTGCGTTCT	
pET-28a-F	TAATACGACTCACTATAGGG	This study
pET-28a-R	GCTAGTTATTGCTCAGCGG	
fedF-3493-F	CCGGAATTCACTCTACAAGTAGACAAGTCTGTT	This study
fedF-3493-R	CCCAAGCTTCTGTATCTCGAAAACAAT	
stx2eA-3493-F	CCGGAATTCCAGGAGTTTACGATAGACT	This study
stx2eA-3493-R	CCCAAGCTTTTCACCAGTTGTATATAAAGG	

Underlined nucleotides denote enzyme restriction sites.

## Data Availability

Data are contained within the article and Appendix A.

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
