# Peer review of "Oral Immunization with Attenuated Salmonella Choleraesuis Expressing the FedF Antigens Protects Mice against the Shiga-Toxin-Producing Escherichia coli Challenge"

_biomolecules, 2023, doi:10.3390/biom13121726_

Round 1
Reviewer 1 Report
Comments and Suggestions for Authors
In this study Zhang et al. generate two vaccine candidates based on a previously described vaccine vector of S. Cholerasuis (rsC0016). The prototypes are made by heterologous expression of two major STEC virulence factors: a fimbriae adhesin (FedF) and a mutated shiga toxin (rSTx2eA) in a pYA3493 plasmid. Even thought commercial vaccines for swine are available, the goal of this manuscript is the current interest due to the worldwide relevance of piglet edema disease. The manuscript is well written and interesting to read. The objectives and results are presented concise that make the paper follows clearly. The conclusions are supported by the experimental data collected.
The authors show that both vaccine candidates are immunogenic after oral inoculation, and demonstrate substancial levels of IgG (serum), IgA (vaginal lavages), and IL-4 and IFN-g cytokines (spleen supernatants) in samples recovered from vaccinated mice. The protection efficacy was evaluated using a muscle mouse model of infection against one STEC strain (STEC20). Survival rates obtained after immunization with rsC0016(pS-FedF) were superior than with rsC0016(pS-rSTx2eA). Overall, the authors claimed that S. Cholerasuis rsC0016(pS-FedF) is a promise candidate against Shiga toxing-producing E. coli.
Main comments:
I find that using a muscle route for infecting mice in the protection experiments does not make any sense. Which is the rationale? The authors have already been developed an oral model for vaccination which, in my opinion, should be also use for infecting mice to evaluate protection. Only by using the oral infection model it could be mimicked the natural route of infection of STEC. Otherwise the role of virulence factors (which are the Ags selected for vaccine formulations: adhesin and enterotoxin) during infection may be underestimated.
If the authors have the possibility to perform additional animal experiments it would be of high interest to test if co-immunization with both vaccine candidates could improve vaccine efficacy against STEC and to include more diversity of E. coli strains. In addition, the detection of IgG anti-S. Cholerasuis OMPs opens the probability to cross-protect against Salmonella infection, that could have a positive impact to piglet paratyphoid fever. I encourage authors to consider further investigations in this regard.
Lines 112 and 122. Table 1 and Table 2 are missing.
Lines 134-135. It is not clear if mice were administrated with 10 ug or 20 ug each dose.
Line 174. Could authors include a visual immunization schedule for immunization/sample recovering/ challenge? Provide the number of total mice used in the study. It would be desirable to include the number of mice in each experimental group, and to refer this number in the figure legends.
Line 186. Explain how oral immunization was performed: oral gavaged/pipete feeding, with/without antibody treatment and/or fasting period, volume for inoculation …
Line 192. Why the authors evaluated IgA levels in vaginal fluids instead of fecal samples? The latter are more appropiate after oral immunization.
Line 206. Is that referring to determining lethal doses? Table S1 must be mentioned here.
Figure 2C. Which is the western blot with the expression of Stx2eA?
Line 314. The survival rate obtained by rsC0016(pS-Stex2A) seems to be 50 %, as represented in Figure 5.
Reviewer 2 Report
Comments and Suggestions for Authors
Dear authors,
I appreciate the invitation to review this article. The aim of this article is very important for pig production in the world.
In general, the authors described the study well and the limitations in the discussion.
Regarding the article, I have some questions and suggestions:
Introduction: Many articles used in the introduction describe vaccine use and results for other species. I suggest you describe that the studies are about other species, it's better for the readers.
In the final introduction, the authors present their results. I think it's better to limit the introduction to the purpose of the article and describe the results only in Results and Discussion.
Materials and methods: I didn't find Table 1 and Table 2 in the file. The supplementary material contains other files, such as Table01S and figures. I would like to see the table in text.
Does the STEC20 strain come from sick animals? Is there already a characterization of the strain published? If yes, add references or GenBank number if sequenced.
Was the indirect ELISA used done in house or is it a commercial kit? Was it validated? If yes, add the reference in the text.
Why was the flow cytometry technique not used?
Please review the text in general for nomenclature and italics of bacterial names.
Reviewer 3 Report
Comments and Suggestions for Authors
The current study aimed to develop a Salmonella-based vaccine by constructing a Shiga-like toxin encoding gene into a plasmid vehicle to be expressed in mice. The constructed plasmid has been expressed in stimulated antibodies and protected the mice.
The main concern is that Salmonella is intracellular and when the recombinant protein it could stimulate the production of CD8-t cells besides CD4-T cells. The authors should shed light on this at least in the introduction or discussion.
Comments on the Quality of English Language
ok
Round 2
Reviewer 3 Report
Comments and Suggestions for Authors
The authors replied the raised pointnt and it should be suitable for publication.
Comments on the Quality of English Languageok